# Clinical Effect of Thioglycosides Extracted from White Mustard on Dental Plaque and Gingivitis: Randomized, Single-Blinded Clinical Trial

**DOI:** 10.3390/ijms25105290

**Published:** 2024-05-13

**Authors:** Konrad Michałowski, Aniela Brodzikowska

**Affiliations:** Department of Conservative Dentistry, Medical University of Warsaw, ul. Binieckiego 6, 02-097 Warszawa, Poland; konrad.michalowski@wum.edu.pl

**Keywords:** thioglycosides, white mustard, dental plaque, gingivitis, plant-based products, oral hygiene, toothpaste, dentistry

## Abstract

The antibacterial and anti-inflammatory effect of thioglycosides has already been established. This study investigates the effects of thioglycosides extracted from white mustard, specifically the “Bamberka” variety, in the context of oral hygiene. The aim of the study is to clarify an evidence-based link between the documented antibacterial and anti-inflammatory effects attributed to thioglycosides and their practical application in oral care. A randomized, single-blinded (patient-blinded) clinical study was performed on 66 patients using mustard-based toothpaste for oral hygiene. The patients were examined at baseline and after 6 and 12 months. The values of the Approximal Plaque Index (API), the Plaque Index (PI), and Bleeding on probing (BOP) were taken into consideration. The results show a significant reduction in plaque accumulation, especially after 6 months of using mustard-based toothpaste in all examined parameters. This suggests that thioglycosides from mustard contribute to a considerable decrease in dental plaque accumulation, confirming their potential in natural oral care solutions, which is indicated in the main conclusions or interpretations.

## 1. Introduction

Dental plaque is considered the main etiologic agent in the indication of caries, gingivitis, and its progress to periodontitis. Gingivitis is an inflammatory condition of the gingiva characterized by edema, redness, and bleeding upon probing [1]. Mechanical removal of dental plaque is generally acknowledged as an effective measure for controlling the progression of dental caries and periodontal diseases [2].

Dental plaque biofilm with the composition of specific bacteria species enhances the inflammation process, which is responsible for the development of periodontal diseases. The relationship between the presence of dental plaque and periodontitis was a main figure of periodontal consensuses leading to the new Classification of Periodontal and Peri-Implant Diseases and Conditions (2017) [3].

Periodontitis, a chronic inflammation of tooth-supporting structures, is a multifactorial disease. Severe periodontitis affects 10–15% of the population, and it is the main reason for tooth loss and the sixth most prevalent condition worldwide [4]. Periodontitis is the result of nontreated gingival inflammation connected to bacterial plaque accumulation. Favorable factors causing the disease are smoking [5], immunosuppression [6], diabetes [7], and genetic polymorphism of genes related to the production of inflammatory cytokines and the alteration of leucocytes [6].

The definition of healthy gingiva has also been reconsidered. It now focuses on the absence of visual signs of inflammation and bleeding [8]. The therapeutic approach is mainly focused on removing biofilm, which consists of bacteria, which is mostly covered by home oral care [6,7]. The gold standard in non-surgical periodontal treatment is scaling and root planning (SRP) [9]. It is mainly focused on the manual removal of biofilm and calculous and smoothening of the root surfaces. 

To improve outcomes and avoid bacteria recolonization, ancillary therapies like probiotics [10], chlorohexidine, photodynamic treatment [11], and ozone application [12] have been considered. Many plant additions have also been recently evaluated. In the dynamic growth of plant-based oral care products, a diverse array of natural ingredients has played a prominent role [3]. Among these, plants like Neem, Miswak, Aloe Vera, Rheum palmatum, and Rhamnus frangula stand out for their unique salutogenic properties [10].

The expanding realm of plant-based tooth products shows a rich variety of botanical ingredients. Neem, esteemed for its potent antimicrobial properties derived from compounds like nimbin and azadirachtin, acts as a natural shield against oral bacteria [13,14,15,16,17,18,19,20]. Miswak, sourced from the Salvadora persica tree, has been traditionally revered for its natural dental hygiene benefits, attributed to its silica content and alkaloids [20,21,22,23,24]. Aloe Vera, known for its soothing and anti-inflammatory properties, finds application in oral care formulations to alleviate discomfort and promote healing. Rheum palmatum, a herb with roots in traditional medicine, has been associated with potential antioxidant and anti-inflammatory effects [25]. Rhamnus frangula, derived from the buckthorn plant, adds to the holistic approach of plant-based oral care with its potential as a herbal remedy [26,27,28].

The relationship between a decrease in gingivitis and plant-based resources like postbiotics and *Aloe Barbadensis* leaf juice is constantly being investigated [29]. Nevertheless, a knowledge gap in how to use thioglycosides in dentistry still exists. 

Thioglycosides, as presented in Figure 1, are a type of chemical compound characterized by the presence of a sulfur atom within the glycosidic bond. Structurally, they are glycosides in which the oxygen atom in the glycosidic linkage is replaced by a sulfur atom. Glycosides, in general, are compounds in which a sugar molecule (glycone) is bound to a non-sugar molecule (aglycone) via a glycosidic bond. In the case of thioglycosides, the aglycone portion of the molecule is often a sulfur-containing compound. The sulfur atom introduces unique chemical and biological properties to thioglycosides, and these compounds are found in various plants across different families [30,31].

Thioglycosides are a group of bioactive compounds renowned for their potential health benefits and are abundantly present in various plants that contribute to their overall health-promoting profile. They are present in various plant sources, mainly Cruciferae and Brassicale including Broccoli, Horseradish, Cauliflower, Brussels Sprout, and Mustard. Scientific investigations into thioglycosides have revealed compelling evidence of their antibacterial and anti-inflammatory effects [32]. These compounds might exhibit the potential to inhibit the growth of oral bacteria, including Streptococcus mutans, and mitigate inflammatory processes in the oral cavity, but there is no evidence in clinical studies. Understanding these effects is crucial as we explore the application of thioglycosides in natural oral care solutions [33,34].

One well-known example of a thioglycoside is allyl isothiocyanate, which is produced by the breakdown of substances and is commonly found in plants like mustard. Allyl isothiocyanate is responsible for the pungent flavor in mustard and exhibits antimicrobial properties, contributing to its traditional use as a preservative [35,36]. The chemical process of transition of thioglycoside into allyl isothiocyanate is described in Figure 2. 

These bioactive compounds, present in the mustard plant, have sparked scientific interest due to their potential health benefits. It has been known since prehistoric times that they possess a high level of bioactive ingredients. The most common species are white mustard (*Sinapsis alba* L.), black mustard (*Brassica nigra* L.), brown mustard (*Brassica juncea* L.), Ethiopian mustard (Brassica carinata A. Braun), rocket (*Brassica eruca* L.), and wild mustard (*Sinapsis arvensis* L.) [37]. White mustard, a common culinary ingredient and a plant readily available in our European climate, traditionally valued for its distinctive flavor, adds an intriguing dimension to our study [38,39,40,41].

In recent years, Piętka et al. presented a new variety of zero-erucic white mustard called “Bamberka” [42]. The major benefit of this new variety is the reduction in the level of erucic acid in mustard oil, which has negative health implications [43]. The European Food Safety Authority (EFSA) states that the tolerable daily intake of erucic acid is 7 mg/kg body weight [43]. The next most important factor is the high concentration of thioglycosides in the ‘Bamberka’ variety. For a long time, white mustard has been used to produce spices, including mustard, as well as in herbal medicine in the form of an extract from mustard seeds (whole or fragmented) [44]. Aqueous extracts contain slimes and thioglycosides. They are used orally as a protectivum. White mustard seeds contain thioglycosides, mainly sinalbin, as presented in Figure 3. 

Seed fragmentation leads to myrosinase enzyme release from other parts of seeds. Myrosinase leads to a cyclic transition of thioglycosides to their derivatives, which are fat-soluble [45]. These compounds are responsible for mustard’s pungent taste.

Previous studies have mainly focused on the derivatives of thioglycosides obtained from the mustard plant [46].

There is a knowledge gap in the use of mustard-based products in oral healthcare products. No correlation has been found in the literature. There has been no clinical study disclosing the effect of mustard on periodontal health. Our study is the first clinical trial to fill this gap in science. 

The aim of this study was to evaluate the effect of thioglycosides extracted from white mustard “Bamberka” on dental plaque and gingival inflammation. This study aims to establish a concrete and evidence-based link between the documented antibacterial [47,48,49] and anti-inflammatory effects attributed to thioglycosides and their practical application in oral care. By focusing on patients using mustard-based toothpaste, we aim to elucidate how this natural component, rich in thioglycosides, may play an essential role in promoting oral health. This exploration may develop a bridge between traditional plant-based knowledge and modern oral care practices regarding mustard and its bioactive compounds using evidence-based procedures.

## 2. Results

Initially, 149 patients were screened, 16 were excluded, 133 were enrolled in the study, 66 patients were allocated to the intervention with toothpaste containing thioglucosides, and 67 patients were allocated to the control group—use of toothpaste without thioglucosides. Four patients were lost during follow-up. A flow chart of the study is shown as a CONSORT flow diagram in Figure 4.

The results for PI, API, and BoP divided into groups and study periods (T_0_, T_1_, and T_2_) are charted and presented in Table 1. Plaque Index and API Index measurements are shown in Figure 5 and Figure 6. 

A significant change was observed in each examined group, but the most significant reduction in plaque accumulation was observed after 6 months, especially in the high-DMFT-PD(−) and high-DMFT-PD(+) study groups. The difference between 6 months (T_1_) and 12 months (T_2_) indicated a progression, but not as rapid. The results of intra and intergroup comparisons using ANOVA one-way analysis of variance with repeated measures showed that there was a significant difference between the variables, F = 44.34, *p* < 0.05. Descriptive statistics are presented in Table 1.

### 2.1. Plaque Index (PI) Measurement Results

On the basis of clinical assessment, no significant differences were found between the groups. Mean PI values were lowest in the low-DMFT-PD(−) group and high-DMFT-PD(−) group, amounting to 3.38 in the low-DMFT-PD(−) group and 3.78 in the high-DMFT- PD(−) group, respectively. The highest mean PI level was observed in the high-DMFT-PD(+) group (PI = 4.35). The mean PI value in the control group in the initial examination was 4.05. 

PI levels changed dynamically over 6 and 12 months in the test groups, presenting a downward trend. The observed changes in numerical values of the Plaque Index were statistically significant. The biggest decrease was observed in the high-DMFT-PD(−) and high-DMFT-PD(+) groups. In the final examination, in the high DMFT-PD(+) group, the mean PI value (1.42) reached levels similar to the low-DMFT-PD(−) group. The lowest PI level, amounting to 0.76, was observed in the high-DMFT-PD(−) group. The most dynamic changes were observed after the first six months of the trial. In all groups, a statistically significant decrease in the PI value was observed. Over the next six months, the dynamics of the shift decreased. It was the most prominent in the high-DMFT-PD(+) group. In the control groups, a stable decrease in mean PI values could be observed, compared to the experimental groups. In the final examination, the mean values of the PI index in the control groups were significantly lower than the initial values in these groups.

### 2.2. Approximal Plaque Index (API) Results

We observed the lowest API initially in the low-DMFT-PD(−) and high-DMFT-PD(−) groups, while the high-DMFT-PD(+) group showed the highest value. Dynamic decreases in mean API values (−16.6; −19.0; −24.7; −10.2) were observed in all groups after 6 months (T_1_). Then, after 12 months (T_2_), a reduction was still visible but much lower (−4.6; −3.5; −7.0; −4.7). In the high-DMFT-PD(−) and high-DMFT-PD(+) groups, the plaque reduction was the highest. Visualization of the above results is presented in Figure 6.

### 2.3. Bleeding on Probing Index (BoP) Results

The most distinctive values of the number of bleeding sites during the initial probing were observed in the low-DMFT-PD(−) and high-DMFT-PD(+) groups. In the low-DMFT-PD(−) group, the BoP value was lower than in the rest of the groups—with a mean BoP value of 32.90%. Such a low percentage of sites exhibiting bleeding on probing could result from the fact that patients in this group had low DMFT values and were not diagnosed with periodontal disease. The highest mean BoP level, amounting to 60%, was observed in the high-DMFT-PD(+) group. During the follow-up, after six and twelve months, a constant, dynamic decrease in BoP values was observed in all the tested groups, independently of the toothpaste used. The changes were statistically significant. The rate of this BoP mean change was the lowest in the control group. The most dynamic changes were observed in the high-DMFT-PD(−) and high-DMFT-PD(+) groups. The BoP mean values decreased in these groups very fast. After one year, the lowest value, BoP = 20%, was observed in the high-DMFT-PD(−) group. The analysis of these study results revealed that the most significant drop in the BoP mean value was observed in the first 6 months in all experimental groups, but not in the control groups. The decrease was also present over the next 6 months, but the dynamics of the changes were much lower. The changes in BOP parameters are visualized in Figure 7.

## 3. Discussion

Gingivitis caused by periopatogens present in dental plaque is a serious factor causing the loss of tooth-supporting structures. Non-sufficiently treated gingivitis may lead to loosening of the tooth and edentulism. The fact that SRP plays a major role in the treatment of periodontitis is not questionable. Adjuvant therapies that lead to preventing or slowing down bacteria biofilm recolonization should be taken into consideration [50]. The novel role of postbiotics was investigated as complementary to SRP treatment of gingivitis [51]. The advantages of this type of therapy prompted us to search for other natural ingredients that might be beneficial. 

A literature search in databases like PubMed and EMBASE did not show any similar study. The novel concept used, concerning mustard-based compounds in oral hygiene, makes this work unique. This is why we cannot compare it to any other clinical study performed previously. Our randomized, single-blinded clinical trial provides an adequate evidence level to reduce the knowledge gap in this field. 

The high concentration of thioglycosides in the “Bamberka” variant plays an essential role in our reasons for picking this plant for our study. The antibacterial and anti-inflammation properties of thioglycosides extracted from other plants, like *Salvadora persica* [20], were described in previous studies. Nevertheless, mustard as a source of an examined substrate had never been considered as a source of oral health products. Only one in vitro study by Echel et al. [52] showed that oral pathogens are susceptible to mustard oil. The novel variant “Bamberka” provides an opportunity to conduct this kind of research. The geographical dispersion of *Brassicaceae* species and mustard in central Europe directly leads to scientific exploration of their properties. There is a clear relationship between the interest and accessibility [53] of plant products and their use for daily care.

Most of the *Brassica* species contain allergens, but the vast majority are related to mustard. According to the Food and Agriculture Organization of The United Nations, mustard is the most commonly used product to manufacture prepacked food like seasoning and flavoring agents and texture control agents [54]. Verhoeckx et al. [55] precisely described this problem and the implications of mustard seed processing in the presence of allergens. Food allergies limit the use of this compound, but the overall prevalence is relatively low [56,57]. Most of the case reports disclosing this problem were related to food allergies. A contact allergy would cause a significantly low level of complications. Labeling toothpaste as a product only for external use will reduce this potential problem. 

During the study, patients were divided into six groups. Control groups were treated with toothpaste without thioglycosides. Patients were not informed of their allocation to the test or control group. The increase in periodontal health parameters could be a result of regular oral hygiene kept by the patients while undergoing the study and the Hawthorne Effect [58]. It also supports the statement that non-fluoride oral products, especially plant-based home oral care products that contain Neem or Aloe vera extracts, are appropriate to maintain good periodontal status. The highest results of the measured parameters in the initial examination were in the high-DMFT-PD(+) group. The authors would like to determine whether changes in these parameters occurred in addition to the periodontal status of patients. It is crucial that plaque accumulation, a reduction in gingival bleeding, and an improvement of periodontal parameters were observed in all four groups. Initial relationships between the groups remained and flattened after 12 months, but were nevertheless preserved. That clearly justifies the reason for patient group categories. 

Improvement in oral hygiene parameters was significant in all groups, but mostly in the high-DMFT-PD(−) and high-DMFT-PD(+) groups. This relationship was observed in every parameter measured. In each case of comparison, the most relevant reduction was observed in the first 6 months of the trial. In the second period, the trend was still visible but was lower. Hygienic parameter improvement was more significant in periodontally compromised patients in the high-DMFT-PD(+) group than the control group with variable patients and the low-DMFT-PD(−) group containing patients with healthy gingiva. This is a significant indication that thioglycoside-based pastes might be recommended for patients with periodontal diseases. BOP was reduced by 38% in 12 months. All this supports the hypothesis that thioglycosides slow down dental plaque accumulation. Gingiva examination revealed improvements in all groups, with the most significant decrease in mean POB values observed in groups using toothpaste with thioglycosides. 

Mustard, as is commonly known, has a pungent smell and taste. It might not have been acceptable to everyone. Due to the low concentration of mustard oil (5%), the taste was mild, so patients did not complain about the taste of toothpaste their during second and third appointments.

Toothpaste composition must be developed in future studies. Stability and stress tests also have to be performed. The authors suggest applying the Accelerate Stability Assessment Program (ASAP) [59] before the commercial use of this product.

This pioneering study involving a small group of patients cannot be compared to any similar study, because there was no similar research available in the literature. The study validates the prophylactic and therapeutic properties of thioglycosides in reducing gingival inflammation. The promising outcome might be an indication for future research. It is recommended to continue research while taking more parameters into consideration, and longer studies will be necessary to confirm our findings and to better understand how these compounds work. This research contributes to the ongoing exploration of plant-derived ingredients for innovative evidence-based dental and gum care products. It also meets the expectations of the rapidly growing group of patients preferring fluor-free home oral care products. Justification of the use of this kind of product can be found in the improvement of tested parameters in the control group.

## 4. Materials and Methods

The present study obtained a positive affirmation from the institutional review board (KB/58/2011) and was carried out at the Department of Conservative Dentistry of the Medical University of Warsaw. All clinical procedures were achieved in accordance with the Helsinki Declaration of 1975, as revised in Tokyo in 2013. Participants were examined in the outpatient clinic of the University. All patients were informed about the study’s objectives, as well as possible risks and profits of participating in the study.

### 4.1. Mustard Experimental Paste Preparation

Mustard oil was extracted from white mustard “Bramberka” using the Soxhlet reference method. Using the Soxhlet extractor made from BORO 3.3 glass according to norm DIM 12602 (PHU Chemo-lab, Ruda Śląska, Poland), two formulas were simultaneously assessed: toothpaste made from fragmented entire white mustard seeds (analogously to mustard production) and ethanol extract of fragmented mustard seeds. The concentration of main sinalbin derivatives in the obtained toothpaste was measured by means of a UV absorption test (280 nm). Through the use of high-performance liquid chromatography (HPLS) (Shimazu company, Kyoto, Japan) with a UV diode array detector (DAD) (Knauer, Berlin, Germany), the concentration was expressed in relation to the master alcohol extract of 1 g of mustard seeds. 

After the evaporation of alcohol, the remnants (containing mustard oil and thioglycosides transitions’ derivatives) were added to the toothpaste base, which had been prepared in advance. 

The toothpaste base composition was typical for the cosmetic industry [60,61] and is presented in Table 2. The stability of components in toothpaste was checked and confirmed through the use of HPLS methods. 

After enrichment with milled mustard seeds (5%) and alcohol extract (0.3%), a reevaluation of the paste was performed after two weeks with HPLS methods showing the time stability of the experimental paste composition. 

The control formula was a base toothpaste without mustard seed preparation. Both pastes were packed into plastic tubes and coded to ensure that participants did not know the content. 

### 4.2. Toxicologic Study

In the experimental study, an extract from a food product was used. It has also been used for many years in herbal medicine. The extract preparation process is typical for the food industry. According to the EFSA, the daily intake of harmful erucic acid is 8 mg/kg [43]. A potential food allergy reaction, which was reported in adults, was correlated to the mean cumulative dose–response circa 125 mg of mustard seed [62]. Mustard seeds are responsible for allergy reactions in approximately 1% of children [63]. The external use of toothpaste decreases the risk of potential allergic incident occurrence. Inclusion criteria (aged older than 18) eliminated the child population, which is more susceptible to a mustard allergy [64]. The concentration of the active ingredients in the prepared formulations was lower than in mustard. For these reasons, no preliminary toxicological studies were performed on laboratory animals. 

### 4.3. Sample Calculation

There has been no similar study conducted before. We made estimations for the proper sample size, ensuring the correlations were statistically significant. The sample size calculation (alpha = 0.05; power = 80%) for two independent research groups and a continuous primary endpoint was calculated. We expected to obtain differences in the mean of approximately 20%. A sample size of 66 participants per group fulfills the statistical criteria. The sample was calculated using the Clinical Sample Calculator (Clin Cal Lcc https://clincalc.com/stats/samplesize.aspx accessed on 12 May 2024).

### 4.4. Inclusion Criteria

The inclusion criteria were as follows: the presence of teeth in the mouth, aged 18 years old or over, API and BoP values in the initial examination of more than 20%, regular oral hygiene at home, and motivation to take part in the study. Participants were gathered from a pool of patients submitted to the Outpatient Clinic of the Department of Conservative Dentistry, Medical University of Warsaw. Each of them agreed to participate in the study and signed a written consent form. 

### 4.5. Exclusion Criteria

The exclusion criteria were as follows: systemic disease, smoking, diabetes, long-term medication, pregnancy and nursing, or a declared allergy to mustard. Participants with orthodontic appliances were also excluded. Patients with restorative treatment that started during the study were excluded and considered “lost in the follow-up”.

### 4.6. Clinical Study Design

The study was conducted at the Department of Conservative Dentistry, Medical University of Warsaw, involving 149 participants (81 males and 68 females) aged 28 to 62 years. Patients were screened according to the exclusion criteria. Sixteen patients were removed, while 133 were enrolled in the study. Initially, each patient was evaluated through medical history taking while completing a survey related to systematic diseases and allergies. Each patient underwent a preliminary clinical examination according to generally accepted principles, and the required conservative treatment and hygienic protocol were provided prior to the study to avoid outcome disturbances. Baseline appointments consisted of a dental examination involving the calculation of the DMFT index (decayed, missing, and filled index) and the API index (approximal plaque index), CPI (community periodontal index) calculation, and an oral hygiene evaluation including PI (Plaque index), BOP (Bleeding on probing), and gingiva observation [65]. 

The community periodontal index (CPI) is the result of the development of the Community periodontal index of treatment needs (CPITN) by changes in the World Health Organization oral health survey [66]. The mouth of a patient is divided into sextants, and ten teeth are taken into consideration, i.e., 17, 16, 11, 26, 27, 37, 36, 31, 46, and 47. The examination involves evaluating the presence of sub- and supragingival dental calculus, the occurrence of gingival bleeding, and the measurement of periodontal pockets with probing depths between 3.5 and 6.0 mm. The examination is performed using a periodontal probe with a 0.5 mm ball tip. The probe has black band markers at 3.5, 5.5, 8.5, and 11.5 mm and is called the WHO probe. Probing is performed with force not exceeding 20 g [67]. The results are marked in each sextant as follows: 0—healthy no bleeding.1—bleeding visible after probing. 2—calculus present during examination, but all of the black bands are visible on the probe.3—4–5 mm pocket (gingival margin within the black band on the probe).4—pocket 6 mm or more (the black band on the probe is not visible).

The Decayed, Missing, and Filled Teeth (DMFT) index is the predominant population-based measure of caries experiences worldwide. This index gives the sum of an individual’s decayed, missing, and filled permanent teeth or surfaces (DMFS) [68].

PI was measured according to Loe’s criteria. It means that during clinical examination, plaque was detected on the gingival margin and scored as follows: 0—if no plaque; 1—a thin layer of plaque at the gingival margin only detected by scraping with a probe; 2—a moderate accumulation of plaque within the gingival pocket, and plaque is visible with the naked eye; 3—plaque presence around the gingival margin with the vast majority of interdental spaces filled with plaque [69].

The Approximal Plaque Index (API) according to Lange et al. [69] is related to a patient’s oral hygiene status. The buccal side of the first and third quadrants and the lingual/palatal side of the second and fourth quadrants are examined. Each positive plaque finding is noted, and the sum of the total positive findings is used to calculate API using the following formula: (sum of positive findings/sum of investigated approximal spaces) × 100%. API is a simple numerical grading method of patients’ oral hygiene: an API value below 39% represents optimal oral hygiene and a value above 40% indicates insufficient oral hygiene [70,71].

Bleeding on Probing (BoP) during the periodontal examination is directly related to the inflammation process in the gums. The examination was performed using the periodontal probe recommended by the WHO. After probing sockets, the percentage of bleeding sockets will indicate the level of inflammation [72,73].

After baseline data collection, the patient pool was divided into six groups regarding the following key. Participants were categorized based on their DMFT and CPI values into 3 groups, which were then simply randomized using an online randomization tool (http://www.randomization.com accessed on 20 June 2023) into experimental and control groups. This resulted in 3 groups allocated to use the enriched experimental paste and 3 control groups:

Group I: Low DMFT, no periodontal disease (CPI = 0) titled “low DMFT-PD(−)”

Group II: High DMFT, no periodontal disease (CPI = 0) titled “high DMFT-PD(−)”

Group III: High DMFT, periodontal disease present (CPI = 1, 2 or 3) titled “high DMFT-PD(+)”

Group IV: Low DMFT, no periodontal disease (CPI = 0) titled “control group—low DMFT-PD(−)”

Group V: High DMFT, no periodontal disease (CPI = 0) titled “control group—high DMFT-PD(−)”

Group VI: High DMFT, periodontal disease present (CPI = 1, 2 or 3) titled “control group—high DMFT-PD(+)”

Patients in the low-DMFT-PD(−), high-DMFT-PD(−), and high-DMFT-PD(+) groups were allocated to experimental toothpaste containing thioglycosides. Control groups received paste with a control formula—without thioglycosides. The study was simple-blinded (patient-blinded), so all toothpaste samples were delivered in the same non-marked plastic tube so that the patient did not know their allocation. Participants were asked to follow home routine oral hygiene procedures, namely brushing two times a day for 2 min with the experimental product. Participants were instructed not to change any daily routine or hygienic behavior. 

The study took 12 months, with assessments conducted before treatment initiation (T_0_), at 6 months (T_1_), and at 12 months (T_2_). Professional hygienization was performed during the preliminary examination to achieve the most relevant study outcome. Dental treatment during the study was provided according to patients’ needs but it was an exclusion criterion for the study. Key measurements taken during the study were as follows: the Silness–Loe Plaque Index (PI) [74], the Approximal plaque index (API) according to Lange et al. [69], and Bleeding on Probing (BoP) according to Ainamo and Bay [75]. Each of them was evaluated at each time point: at baseline (T_0_) in the preliminary examination, and then after 6 months (T_1_) and 12 months (T_2_). Additionally, a comprehensive dental and periodontal examination was performed during each time point to eliminate any adversities present. 

### 4.7. Statistical Analysis

The statistical analysis was performed using Statistica v. 13 (TIBCO Software Inc., Palo Alto, Santa Clara, CA, USA). A test of normality was conducted using the Shapiro–Wilk test. Descriptive statistics, including the mean, standard deviation, and the number of subjects, were employed. Parametric t-tests for paired samples to compare each group were used for data analysis. Significance levels were set at *p* < 0.05.

## 5. Conclusions

Thioglycosydes extracted from white mustard had a significant effect on oral hygiene and periodontal health.

A significant reduction in plaque accumulation and gingivitis was observed, especially after 6 months in the three groups that were allocated mustard-based toothpaste. This suggests that the inclusion of thioglycosides from mustard in oral care formulations contributes to a notable decrease in dental plaque. The findings suggest that mustard-based toothpaste enriched with thioglycosides could be a valuable addition to natural oral care solutions. However, the evidence is not sufficient, so further analyses in randomized clinical trials must be performed. 

## Figures and Tables

**Figure 1 ijms-25-05290-f001:**
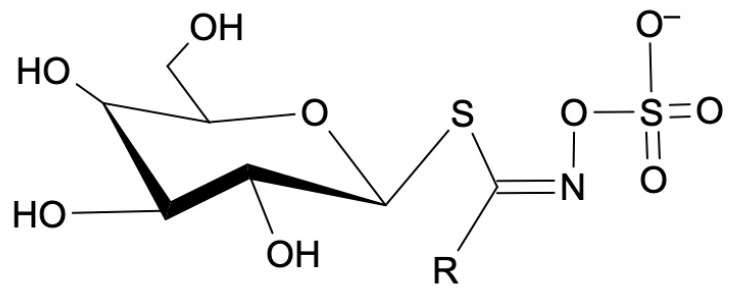
General molecular formula of thioglycosides. R—amino acid group.

**Figure 2 ijms-25-05290-f002:**
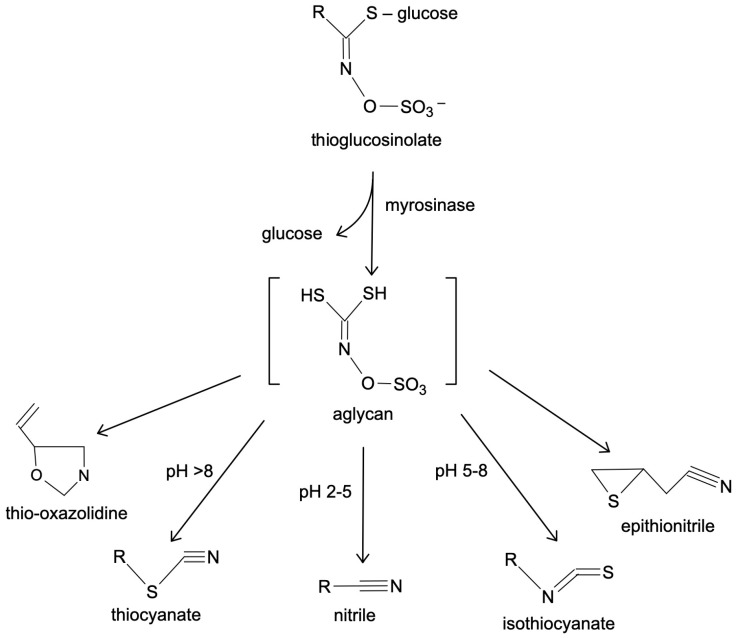
Thioglycosides’ transitions.

**Figure 3 ijms-25-05290-f003:**
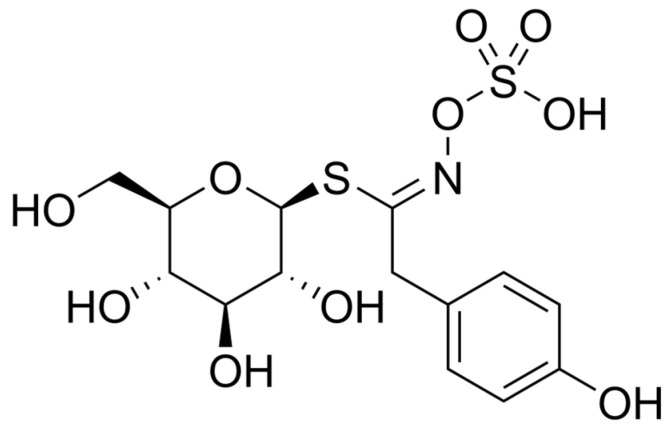
Sinalbin, the main thioglycoside of white mustard.

**Figure 4 ijms-25-05290-f004:**
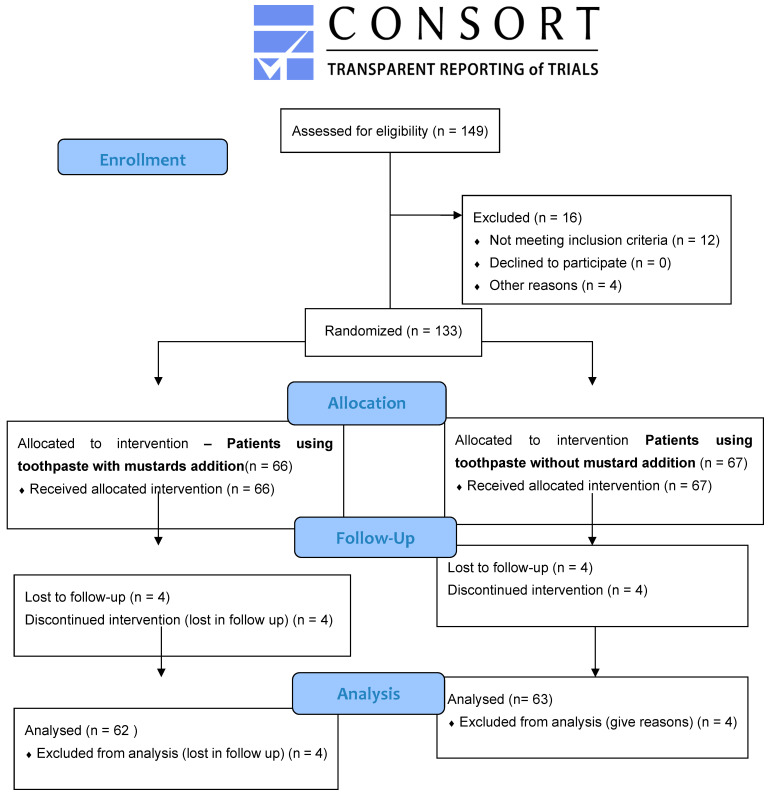
CONSORT diagram showing the study outline.

**Figure 5 ijms-25-05290-f005:**
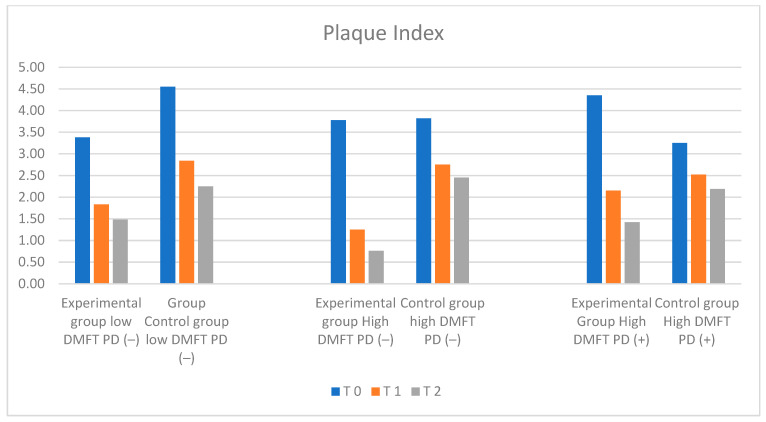
Illustration of changes in PI in each group during the study.

**Figure 6 ijms-25-05290-f006:**
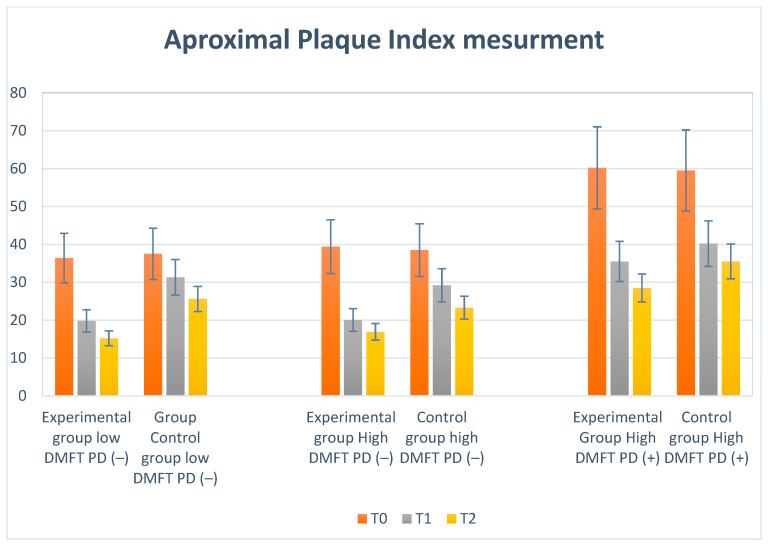
Illustration of changes in API in each group during the study.

**Figure 7 ijms-25-05290-f007:**
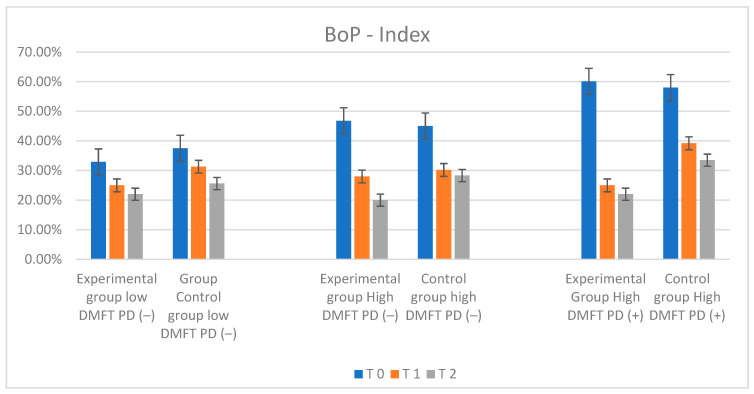
Illustration of changes in BOP index in each group during the study.

**Table 1 ijms-25-05290-t001:** Clinical parameters: PI, API, and BoP at baseline (T_0_), 6 months (T_1_), and 12 months (T_2_).

Group	N	Time	API Mean (SD)	Significance *	PI Mean (SD)	Significance *	BOP Mean (SD)	Significance *
low DMFT-PD (−)	22	T_0_	36.4 (19.5)	A	3.38 (1.13)	a	32.9 (10.2)	*A*
	22	T_1_	19.8 (15.2)	B	1.83 (0.42)	b	25.0 (6.0)	*B*
	20	T_2_	15.2 (13.6)	C	1.48 (0.33)	b	22.0 (5.0)	*C*
high DMFT-PD (−)	19	T_0_	39.4 (20.6)	D	3.78 (0.72)	c	46.8 (8.4)	*D*
	19	T1	20.4 (16.6)	B	1.25 (0.52)	d	28.0 (12.0)	*E*
	19	T_2_	16.9 (12.3)	C	0.76 (0.48)	e	20.0 (7.0)	*F*
high DMFT-PD (+)	25	T_0_	60.2 (35.2)	E	4.35 (0.58)	f	60.1 (12.2)	*G*
	23	T_1_	35.5 (19.5)	A	2.15 (0.76)	g	25.0 (6.2)	*B*
	23	T_2_	28.5 (16.8)	G	1.42 (0.82)	b	22.0 (5.5)	*C*
Control group low DMFT-PD (−)	22	T_0_	37.5 (22.6)	A	3.25 (1.24)	a	31.2 (10.1)	*A*
	20	T_1_	31.3 (15.1)	G	2.52 (1.09)	h	28.0 (9.0)	*E*
	20	T_2_	25.6 (12.8)	F	2.19 (0.89)	g	26.0 (10.1)	*B*
Control group high DMFT-PD (−)	20	T_0_	38.5 (25.9)	D	3.82 (1.13)	c	45.0 (8.1)	*D*
	20	T_1_	29.2 (12.1)	G	2.75 (1.15)	h	30.2 (15.2)	*E*
	19	T_2_	23.3 (9.5)	F	2.45 (1.04)	h	28.3 (12.6)	*H*
Control group high DMFT-PD (+)	25	T_0_	59.5 (19.6)	E	4.55 (0.98)	f	58.0 (15.4)	*G*
	25	T_1_	40.2 (12.3)	H	2.84 (1.19)	h	39.2 (20.3)	*I*
	25	T_2_	35.5 (20.2)	I	2.25 (1.32)	h	33.5 (18.2)	*J*

DMFT—Decayed, Missing, Filled Tooth index, PD—Periodontal Disease N—Number of patients, T_0,1,2_—Time periods of study, API—Approximal Plaque Index, PI—Plaque Index, BOP—Bleeding on probing, SD—Standard Deviation, * significant intragroup and intergroup differences assessed using Dunn’s post hoc test. * The means with the same letters (capital, small, or italics) are not significantly different (*p* > 0.05).

**Table 2 ijms-25-05290-t002:** Composition of toothpaste used in the study.

Material	Content (%)
dicalcium phosphate dihydrate	38
demineralized water	32.4
glycerol	25
silica (silicon dioxide)	2.4
carboxymethyl cellulose	1.2
sodium lauryl sulphate	0.6
sodium benzoate	0.2
sodium methyl hydroxybenzoate	0.2

Toothpaste base was enriched with milled mustard seeds—5% of the mass; ethanol extract—the remnants obtained after alcohol evaporation (0.3%).

## Data Availability

The data presented in this study are available upon request from the corresponding author.

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
