# Peer review of "Clinical Effect of Thioglycosides Extracted from White Mustard on Dental Plaque and Gingivitis: Randomized, Single-Blinded Clinical Trial"

_ijms, 2024, doi:10.3390/ijms25105290_

Round 1

Reviewer 1 Report (New Reviewer)

Comments and Suggestions for Authors

Dear authors, thank you for a nice presentation of your work.

The introduction and methods as well as the information included are well presented. It would be preferable though to work on your two first paragraphs of the introduction and make them stronger with more evidence-based material and citations. We lack citations at the beginning of your article. then lines 64-66 do not make any sense. Perhaps you consider correcting the meaning. The part about "toxicology" needs also a better explanation and citations. please also revise lines 175-178. The discussion part starts well but needs to be stronger since your work is unique and you do not show this well enough. You also present a methodology quite clearly and evidence-based so in the discussion you need to write also something about it. References need to be corrected according to guidelines for authors

Fair enough your approach. Keep up the good work!

The reviewer

Comments on the Quality of English Language

minor corrections needed

Author Response

Dear Reviewer,

We are very thankful for your positive response for our work.

Regarding to your suggestion we add more references related to periodontal disease and mustard as a plant. We corrected the marked lines to make them more clear for the reader.

Toxicology part is developed now with references, we also discuss and include the problem of allergy to mustard. Mentioned lines 175-178 are referred to sample size calculation. We stated that there were no previous study like that so we cannot compare it. We calculated the size due to statistical estimations. It is now add in this paragraph.

References are now corrected due to the MDPI Guidelines for authors.

We hope that you will enjoy the corrected version.

Best Regards,

Authors

Reviewer 2 Report (New Reviewer)

Comments and Suggestions for Authors

Dear Authors, 

my comments are listed below: 

-full stop should be written after the references in brackets;

-mark the figures and tables, which are not, in the text;

-where the paients randomly allocated? You should define was this blinded or double blinded study and add it in the main text and the Title;

-part 2.3: according to which data You calculated sample size? You should add reference or previous study that You used for calculation;

-part 2.7.: You used parametric and non parametric test which is not correct. You should, depending on the result of distribution test, use parametric or non-parametric tests (not both):

- add which statistical test You used and p-values below the Tables and Figures;

- Why You did not use intergroup comparison?;

-line 429: 4 groups - this is the opposite to to what You presented previously in the Material and Methods and Results parts. You had 6 groups. This is confusing. 

- as is known that the mustard is one of the most common allergen, You should add about that in the Discussion part (potencial complications)

Comments on the Quality of English Language

*

Author Response

Dear Reviewer,

Thank you for your suggestions. We highlighted the points you mentioned and gave clear answer to every point. 

-full stop should be written after the references in brackets;

We replace it now in the correct order due to Guidelines for Authors by MDPI.

-mark the figures and tables, which are not, in the text;

Done. All figures and tables are marked in the text now.

-where the patients randomly allocated? You should define was this blinded or double blinded study and add it in the main text and the Title;

Patient were randomly allocated, the study was single blinded – patient blinded. Patient pool was divided into three groups due to periodontal parameters. Then each group was randomized in to two experimental and control. We did simple randomization with the online tool. Now its clarified in text and proper reference is also included.

-part 2.3: according to which data You calculated sample size? You should add reference or previous study that You used for calculation;

The importance of sample size calculation was implemented in the planning of this project. The importance was significant for the group with more than 60 records. The plan was to conduct research study to establish the beneficial effects to draw a precise and accurate conclusion only with an appropriate sample size. The obtained results indicate that the available sample size was sufficient. Postive outcome of Shaphiro-Wilk test supports the validate the normal distribution of

-part 2.7.: You used parametric and non parametric test which is not correct. You should, depending on the result of distribution test, use parametric or non-parametric tests (not both):

We assume to use parametric test due to the normal distribution.

- add which statistical test You used and p-values below the Tables and Figures; - Why You did not use intergroup comparison?;

We add the Shapiro-Wilk values in tables and figures. We didn’t use intergroup comparison due to clear and significant reduction of values presented in charts and table.

-line 429: 4 groups - this is the opposite to to what You presented previously in the Material and Methods and Results parts. You had 6 groups. This is confusing.

This mistake was done due to editorial flaws. It is clear and correct now.

- as is known that the mustard is one of the most common allergen, You should add about that in the Discussion part (potencial complications)

Thank you for mentioning this problem. Every patient submitted for the trial was asked for general health and allergies. All participants decline allergy to mustard. Using a oral-care product with mustard may lead to contact allergy not food allergy. However it might provide a potential complication, so we include it. Additional paragraph in discussion part with proper references is now present in the text. Proper explanation and reference was added in ‘toxicology’ part.

Best regards,

Authors

Round 2

Reviewer 2 Report (New Reviewer)

Comments and Suggestions for Authors

Dear Authors, 

in Table 2. please add p-valus for each variable and group in separate column for each (not in the row like you did now). You should point statistical difference between T0 and T1, and T1 and T2. 

Author Response

Dear Reviewer,

Thank you for your review and highlighting the inaccuracy in the resolute presentation.

We are submitting the corrected text. Regarding to missing  intra and intergroup comparation. They are  presented now in columns in table 2. We used post hoc test to compare the resoults in T0, T1, T2 time. Comparations are marked using capital, small and itallic letters to differentiate compared parameters. The values with the same letters are not sigificantly diferent (p> 0.05). In case of any suggestions, we are open to present the resoults in a diferent way. We strongly believe you will find in now suitable for publication.

Best regards,

Authors

This manuscript is a resubmission of an earlier submission. The following is a list of the peer review reports and author responses from that submission.

Round 1

Reviewer 1 Report

Comments and Suggestions for Authors

Remarkable manuscript

interest in the dental sector, requires a major revision before proceeding with publication.

Abstract: to highlight the results obtained more and above all the API BOP and PI evaluation indices must be written in full when they appear for the first time in the text.

Keywords: add more, there are few

Introduction: add first of all the concept of health and gingivitis based on the new classification of periodontal disease, and all the natural substances used to reduce inflammatory indices as already studied for some time in the research group of Prof Scribante et al.

Materials and methods: how was the sample size calculated?

In the inclusion criteria, at least the % of bleeding is missing, only 20 teeth do not seem in line with the objectives of the research.

Results: very confusing, requires reorganizing graphs and tables with correct statistical analysis.

Discussion: include among future objectives the possibility of comparing them with postbiotics derived from food waste to evaluate their effectiveness, Butera et al.

Have you focused on the taste of mustard? Which might not please everyone?

Conclusions: modify them according to the changes in the text.

Bibliography: add required references

Author Response

Dear Reviewer,

Thank you, for your kind attention while reading the manuscript. We are grateful for your suggestions what highlighted important facts related to the topic. All of them were included in manuscript. We are thankful for sharing with us works of prof Scribante and Butera that let us improve the discussion and give us more perspective look to the topic.

We would like to respond regarding details,

Abstract and Keywords:

Abbreviations like API, BOP, PI are now placed in full text. 

More Keywords, like “plant-based products”, “periodontitis”, “periodontal disease” are now present.

Introduction:

We include, as suggested the definition of periodontitis based on 2018 European Periodontal Consensus and brief details related to indication and treatment of periodontal disease.

Materials and methods:

We would like to state that this is a pilot study. No statistical trial sample calculation was taken.

Results:

We change the tables and figures to make it more clear and add p values and SD.

Discussion:

We enlarge the discussion, including the suggested topic of Butera et al.  following your recommendations.

Conclusion:

Please find developed conclusion in relation to updated results.

Thank you for suggestions.

Reviewer 2 Report

Comments and Suggestions for Authors

Dear authors and editor,

I write to you in regards to the manuscript entitled “Clinical assessment of the antibacterial and anti-inflammatory effect of thioglycosides extracted from white mustard on oral hygiene”. 

There is an interesting topic being investigated in this study, but the conductance of the study and its report are not adequate as it currently is. There are important information lacking and it is not clear what is the question the authors are trying to respond. Also, the absence of a control group with a conventional toothpaste prevents the comparison of the data, and prevents the readers to infer that the effects (which were not statistically significant) were indeed caused by the thioglycosides in the toothpaste. Lastly, as stated by the authors, “group 4 consisted of patients without any inclusion criteria and was considered as a control group”, so it is safe to point out that the lack of standardization in following the eligibility criteria is a major drawback in this study which compromises its reproducibility. Please note that I am not stating that thioglycosides are not able to indeed improve oral health. I am stating that, in its current form, this study design was not able to measure that. Therefore, I suggest rejection.

Title

As antibacterial and anti-inflammatory effects are technically not being investigated in this study, I suggest altering the title to something evidencing that what is being investigated is the amount of plaque and health of periodontal tissues.

Abstract

The introduction, methods, results and conclusion are separated by semi-colons (;). Please, replace it by points (.). Also, write API, PI and BOP in full when its first mentioned in the abstract, as this can help the reader to understand what is being studied and can also help to locate the paper in a database, as these can serve as keywords.

Introduction

·      Please, adjust how citations 1 and 2 are displayed in the second paragraph of the introduction section ([1][2] instead of [1,2]).

·      Line 76: please adjust the punctuation.

·      Lines 77-80: This paragraph is not clear, especially because of punctuation. Please, adjust it.

·      It is not clear to this reviewer what is the purpose of this study. The authors state that “thioglycosides are a group of bioactive compounds renowned for their potential health benefits”, but do not specify nor reference the health benefits promoted by these compounds. Also, if “these compounds exhibit a potential to inhibit growth of oral bacteria, including Streptococcus mutans, and mitigate inflammatory processes in the oral cavity”, there seems to be no novelty for this study. If there are differences, please add what is the knowledge gap it is trying to fill in the current literature and state the purpose of this study in the last paragraph of the introduction section instead of in the methods section.

Material and methods

·      Please, use the CONSORT checklist to properly report this clinical study. Also, I suggest registering this clinical trial in trials.com or any other similar database the authors find appropriate.

·      Please, move the objective of the study from this section to the end of the introduction section.

·      Please, give more details and references on how the seeds were prepared to produce the toothpaste. Also give a reference to the toothpaste composition.

·      Regarding the toxicologic study sub-section, please add more details and references to previous works indicating that higher concentrations of these compounds were not harmful to the same kind of tissue you are using in this study to justify the absence of a toxicologic study prior to the conductance of this clinical study.

·      Inclusion and exclusion criteria: please, give more details on the inclusion and exclusion criteria used in this study. For instance: were people under the age of 18 included in the sample (I only saw this information later in the manuscript)? Were smokers excluded? How many pockets >5mm were necessary to exclude a participant from the study. Please, also give a flowchart of how many patients were screened, excluded and how many were lost during the follow-ups.

·      It is not clear to this reviewer how people with pockets >5mm were excluded but people with periodontits were categorized in groups 3 and 4. Please, give more details.

·      How was the sample size calculated?

·      I believe it would help the readers if, instead of mentioning groups 1, 2, 3 and 4, the authors would come up with an acronym, such as: “-DMFT-PD; +DMFT-PD; +DMFT+PD; Control”.

·      Lines 175 and 196: please, place the citation right after “Lange et. al.” 

·      Line 196: Please, place the citation right after “Ainamo and Bay”.

·      If “Professional hygienisation was performed as needed”, didn’t it bias the outcome of the study?

·      Lines 200-201: this is not necessary to be place in the methods section. It duplicated the information stated in the results section.

·      Please add the p value for the Shapiro-Wilk test.

Results

·      Please, add the p values and the standard deviation values for the comparison in all figures.

Conclusion

Considering that there is not a group without thioglycosides, it is not safe to conclude that its effect was due to the presence of thioglycosides. It could be from the instruction provided by the investigators in the beginning of the study, from the “professional hygienisation performed as needed”, or even by a Hawthorne effect as these participants knew they were being investigated.

References

Please, adhere to the journal’s instructions for authors.

Author Response

Dear Reviewer,

Thank you, for your kind attention while reading the manuscript. We are grateful for your suggestions what highlighted important facts related to the topic. All of them were included in manuscript. We reevaluated and improved the major topic of our study, to highlight the proper aspects of study. In general, we decided to change title, to clarify proper aim of the study. Investigation the influence of experimental toothpaste on reduction in periodontal health parameters. Also we improve the description of whole study design to make it more simple and clear for the reader.  

We would like to respond regarding details,

Title has been changed as suggested

Abstract :

Abbreviations like API, BOP, PI are now placed in full text. 

Introduction:

Citation adjusted according to MDPI requirements.

We include as suggested, the reference to thioglucosides as a antibacterial agents. We defined the knowledge gap. The were no study linking the mustard-based products in oral health care. That’s make this study novel.  We made  a proper correction in texts following your suggestions.

Materials and methods.

CONSORT diagram is now included. We would like to state that this is a pilot study. No statistical trial sample calculation was taken. We enlarge the paste preparation section including proper literature references.

We reevaluate the inclusion criteria to make them more transparent, they are placed in manuscript. Flow chart about lost patient in study is written and included in CONSORT diagram.

There were no sample size calculation, we considered this study as a pilot study. 

We changed names of the groups, following your suggestions. The groups are now “low DMFT-PD(-)”, “high DMFT-PD(-)”, “high DMFT-PD(+)”, “control group”.

We mention previously in the text about the control group and control paste formula. Control group had various DMFT but also was allocated with control formula of toothpaste, without mustard seed extract.

We also state that hygienisation procedures was done only during first appointment.

We added  p value for Shapiro-Wilk test.

Results.

We change the tables and figures to make it more clear and add p values and SD.

Conclusion.

Please find developed conclusion in relation to updated results.

Thank you for suggestions.

Round 2

Reviewer 1 Report

Comments and Suggestions for Authors

The manuscript has been properly revised and can be published.

Author Response

Thank you for your review and acceptance.

Reviewer 2 Report

Comments and Suggestions for Authors

Dear editor and authors,

I write to you in regards to the revised version of the manuscript now entitled “Clinical effect of thioglycosides extracted from white mustard on dental plaque and gingivitis.” Once again, due to serious methodological flaws I consider this manuscript not appropriate for publication in this journal.

·      Line 47. I believe “with” should be “which”

·      Please correct the paragraph in lines 69-70.

·      Introduction: it is still not clear to this reviewer the knowledge gap this study is trying to fill. For instance, how there is a knowledge gap in the use of mustard-based products in oral healthcare (line 119) if there is evidence that thioglycosides exhibit a potential to inhibit growth of ORAL bacteria, including Streptococcus mutans, and mitigate inflammatory processes IN THE ORAL CAVITY (lines 88-90), especially considering that the title of the manuscript is on the “effect of thioglycosides extracted from white mustard”. If there is not clinical study investigating this effect, then the authors should state that this is the first study to do it clinically. This is not clear in the introduction section.

·      Line 159-160: for how long was the stability test conducted?

·      Line 156-157: were the milled seeds and the ethanol extract inserted in the same toothpaste or were two different experimental toothpastes created?

Sample calculation: I understand this is a pilot study, so it is safe to assume this is an unfinished study and the authors will develop this project further. However, one cannot conclude there is any effect if the number of people included in the sample is not sufficient. Therefore I suggest submitting the study for publication only after finishing it. Otherwise we cannot conclude anything except that there might be a benefit of using a mustard-based toothpaste, which might not even be true.

Exclusion criteria: Line 188: it is not clear what is “restive treatment”. Should it be “restorative treatment”?

Lines 238-239 – “tilted” should be “titled”

The major concern about this study is the fact that group 4 cannot serve as a control group for this study design. Group 4 is not comparable to the other groups because it is composed by very different participants (“variable” is the term used by the authors). Moreover, this group did not use the same toothpaste as the other groups. Basically, the design I think would be appropriate to this study is developing a study with two groups (both composed by people with periodontitis, or by gingivitis), and make one brush with the mustard-based toothpaste and the other with a conventional toothpaste (control group). The design that is being presented in this study does not allow us to conclude that the mustard-based toothpaste is beneficial because: 1) this could have happened because these participants knew they were being examined (Hawthorne effect); 2) the conventional toothpaste is not being used in comparable participants (lack of proper control group); and 3) “professional hygienisation was performed during the preliminary examination” (lines 249-250).

In the CONSORT flowchart, the reasons for exclusion of the participants were not given (please, pay special attention to the “give reasons” between parenthesis).

Author Response

Thank you for the review of the manuscript and all the comments provided. Below, we address the reply for identified issues.

We are grateful for highlighting minor stylistic and language corrections for verses 47, 69, 188, 238 have been made and are included in the manuscript.

Introduction:

There is still insufficient knowledge about the impact of thioglycosides on bacteria in the oral cavity. The present literature only references one in-vitro study illustrating such a relationship. We suspect that thioglycosides will influence periodontal health, but there is a lack of evidence. To solve this scientifical problem we are performing this study. The authors would like to clarify that this is the first clinical study investigating this relation. This has been revised in the text to make it explicitly stated.

Experimental toothpaste composition:

Ground mustard seeds were mixed with alcohol extract in the

experimental paste. This has been clarified in the text for better understanding. The chemical stability of the paste was checked through High-Performance Liquid Chromatography (HPLC) after two weeks. The authors

acknowledge the need to implement the Accelerated Stability Assessment Program

(ASAP) for larger-scale production of the paste. This issue has been addressed in the manuscript, as a new paragraph in discussion section with additional reference.

Returning to the main issue regarding the division into research and control groups, the aim of creating three research groups is motivated by a broader comparison of the impact of the applied toothpaste depending on the gum condition. We have data to recreate a control group only with gum/periodontal inflammation. If we combine all groups "low DMFT-PD(-)," "high DMFT-PD(+)," "high DMFT-PD(+)" into one comparative "variable" group, we will obtain results indicating improvement in all studied parameters compared to the control group. This suggests a result better than the Hawthorne effect and previously performed hygiene – evident in the improvement of studied parameters in the control group. Additionally, we want to note that in each of the three examined groups, the improvement in parameters is greater than in the control group, further supporting this thesis.

Round 3

Reviewer 2 Report

Comments and Suggestions for Authors

Dear editor and authors,

I write to you in regards to the newest version of the manuscript entitled “Clinical effect of thioglycosides extracted from white mustard on dental plaque and gingivitis.” Once again, due to serious methodological flaws I consider this manuscript inappropriate for publication in this journal.

As I previously stated, this is an unfinished study and the authors will develop this project further. The CONSORT flowchart was not properly filled. Please, pay special attention to the “give reasons” between parenthesis. Also, there are several misspellings throughout the manuscript.

More importantly, as it currently is, one cannot conclude that this toothpaste is beneficial because of three main issues:

1 ) the sample size: if the sample size is not sufficient, we cannot conclude that people using this experimental toothpaste would benefit from it, despite any good or bad result that might come from the use of the toothpaste.

2) interpretation of data (main issue): the data does not support the conclusion that the toothpastes are beneficial for any of the participants in this study because of 3 reasons:

·      Reason 1: The control group did not use a mustard-containing toothpaste, but an improvement in their PI index was seen. Thus, the improvement in the PI index of any group cannot be attributed to the use of a mustard-containing toothpaste.

·      Reason 2: The API index value was similar between the control group and the high DMFT-PD(+) group in T0, T1 and T2. Despite using different toothpastes, there were no differences between them, therefore the mustard toothpaste was similar to a conventional toothpaste.

·      Reason 3: The BOP values between groups in T0 were different because this was considered in the allocation criteria. Also, after 12 months, the BOP values for all groups seemed to be similar, but the control group did not brush with the mustard-containing toothpaste, therefore this improvement cannot be attributed to the use of a mustard-containing toothpaste.

3) experimental design (main issue): the control group consisted of patients with “variable DMFT and variable CPI levels”. This is too subjective and biases the interpretation of the data. On top of that, the participants in the so called “control group” did not even brush with the same toothpaste used in the other three groups.